# Optimizing Heat Treatment Conditions for Measuring CFRP and GFRP Resin Impregnation

**DOI:** 10.3390/ma15228182

**Published:** 2022-11-17

**Authors:** Ji Hyun Kim, Bhum Keun Song, Kyoung Jae Min, Jung Chul Choi, Hwa Seong Eun

**Affiliations:** Convergence Research Division, Korea Carbon Industry Promotion Agency, 110-11 Banryong-ro, Jeonju 54853, Republic of Korea

**Keywords:** FRP construction material, carbon fiber reinforced plastic, glass fiber reinforced plastic, thermal gravimetric analysis, unsaturated polyester, pyrolysis

## Abstract

As the use of carbon-fiber-reinforced plastic (CFRP) and glass-fiber-reinforced plastic is frequent in the field of construction, a method for measuring FRP resin content is needed. Herein, thermal gravimetric analysis (TGA) was employed to optimize the heat treatment conditions (temperature and time) for determining the resin content in which only the resin was removed without fiber heat loss. Accordingly, the measurement was performed in 100 °C increments at a resin pyrolysis temperature up to 800 °C with a heat treatment time of 4 h to continuously observe the degree of thermal decomposition of the resin. The thermal decomposition of unsaturated polyester was confirmed at the melting point (350 ℃) regardless of the type of fibers used as reinforcement. In the case of CFRP, most of the resin decomposition occurred at 300 °C. Notably, the resin was removed at a pyrolysis temperature of 400 ℃ and almost no change in weight was observed. However, at a pyrolysis temperature of 500 °C or higher, the thermal decomposition of the fibers occurred partially. The results show that the composite resin was removed within 10 min at a pyrolysis temperature of 400 °C in an air atmosphere when using TGA.

## 1. Introduction

In reinforced concrete structures, microcracks in the structure due to physical impact, aging, fire, corrosion, etc., are the main causes of internal corrosion of steel rebars. Corroded steel rebars cause structural durability and safety problems. In contrast to this, the FRP grid is a composite material with high corrosion resistance and chemical resistance compared to steel rebar [1,2,3,4,5,6,7]. Accordingly, the fiber-reinforced plastics (FRP) grid has received attention as a material that can replace steel rebars. Notably, carbon, glass, aramid, and basalt fibers with excellent mechanical properties are actively applied as fiber reinforcement materials to maximize the advantages of FRP (Table 1) and such fibers have different mechanical properties and fire resistance owing to the differences in the unique bonding of each fiber [8,9,10,11,12].

Carbon fiber has a high tensile strength owing to the strong C=C bonds (sp^2^ bond with a bond strength of 602 kJ/mol). However, it has high brittleness because Young’s modulus is high, owing to a decrease in elongation. The carbon–carbon bond has high fire resistance with excellent thermal stability. Aramid fiber has a low tensile strength owing to the weak bond strength of C-N (bond strength: 305 kJ/mol). However, it has a low Young’s modulus owing to an increase in elongation, resulting in low brittleness compared with carbon fibers. The C-N bond exhibits rupture properties owing to the thermal decomposition at a temperature of 400 °C or higher [13,14]. Glass and basalt fibers have low tensile strength owing to the Si-O-Si bond (bond strength: 452 kJ/mol). However, they have a low Young’s modulus and brittleness. The Si-O-Si bond is thermally stable and has the properties of a heat-resistant fiber. The application of carbon, glass, and basalt fibers as reinforcements in construction is suitable for composites requiring both excellent tensile strength and fire resistance properties.

Meanwhile, the conditions for the resin content analysis of FRP for the FRP construction materials are not presented in the Korea construction standard. When thermal gravimetric analysis (TGA) is employed to analyze the resin content in composites used for FRP construction material, the weight change is observed at a constant heating rate. However, as the pyrolysis is performed at high temperatures (625 ± 20 °C) in an air atmosphere using a muffle furnace, the heat loss occurs not only in the resin content but also in the fiber used as the reinforcement material [15,16]. Therefore, the heat treatment conditions for analyzing the resin content should be optimized without fiber loss. In addition, the morphology and true density of the fiber residue under such heat treatment temperature conditions were analyzed to confirm fiber damage.

In this study, TGA was employed to determine the resin content and optimize the heat treatment conditions (temperature and time) in which only the resin was removed without fiber heat loss. Accordingly, the measurement was performed every 100 °C at a resin pyrolysis temperature up to 800 °C with a heat treatment time of 4 h to continuously observe the degree of thermal decomposition of the resin.

## 2. Experimental and Analysis Methods

### 2.1. Materials

The FRP composites were prepared by hand lay-up method and pultrusion. For the hand lay-up method, unsaturated polyester (UPE, manufacturer: Polynt Composites Korea Co., Ltd., model: G613BT, Wanju-gun, Republic of Korea), hardener (manufacturer: Nouryon, model: BUTANOX M-60), and wax (manufacturer: Polynt Composite Korea Co., Ltd., model: PX-0525) were mixed and used as the composite impregnated resin. Carbon fiber textiles and glass fiber textiles and mat were used for the hand lay-up. Carbon fiber textiles (manufacturer: Sigmatex, plain, 400 g/m^2^, Soest, The Netherlands) were prepared as using carbon fiber (12 K CF, manufacturer: Hyosung, model: H2550, Jeonju-si, Republic of Korea and manufacturer: Toray, model: T-700, Chūō ku, Japan). Glass fiber textiles (manufacturer: OCV, model: WR 570, plain, 570 g/m^2^) and glass fiber mat (manufacturer: OCV, model: M 723, 450 g/m^2^) were used. Aramid textiles (manufacturer: TEI Composites, Chang Hua, Taiwan, model: TI-9013, plain, 170 g/m^2^) were used. A peel ply made of polyester and coated with nylon was used as a release material of the mold and textile fabric in the hand lay-up molding.

### 2.2. Experimental Methods

The FRP composite used for the resin content measurement was prepared directly by the hand lay-up method, which is a molding method for reinforcement lamination and resin impregnation by hand. The hand lay-up manufacturing method involves mold release treatment, mixing resin, inner lamination of peel ply, fiber lamination, and outer lamination of peel ply [17]. To uniformly apply the release agent to the mold (size: 1000 × 1000 mm^2^), a fabric was moistened with the release agent and used to coat the entire mold. The impregnated resin was prepared by mixing the primary material, hardener, and wax in a ratio of 300:6:3 g. To mix the material, the material was put into a 1 L plastic beaker at once and mixed with a stirring speed of 100–150 rpm using a mechanical stirrer at room temperature. A layer of peel ply was laid on the mold, and the impregnated resin solution was poured on it. Then, a roller was used to uniformly impregnate and remove air bubbles. Subsequently, a textile fabric was placed on it, and the resin solution was repeatedly applied to each layer of the textile fabric. A total of four layers of textile fabric were used. The size of the textile fabric and peel ply was the same as the mold size. Finally, the outer layer of the peel ply was added. Curing of the composite was conducted at room temperature for 24 h and a composite thickness of 2 mm was obtained.

### 2.3. Analysis Method

The surface morphologies of the FRP specimens with different pyrolysis temperatures were analyzed using a field emission scanning electron microscope (FE-SEM, manufacturer: Hitachi, Tokyo, Japan, model name: SU8230) at magnifications of 30 and 1000×. The resin content of 10-mg FRP was analyzed using TGA (manufacturer: Dong-il Shimadzu, Kyoto, Japan, model: DSC 204) in the temperature range of 30–800 °C and a heating rate of 5 °C/min in an air atmosphere. The tensile strength of FRP was measured six times using a specimen that had a width and thickness of 25 mm and 2 mm, respectively. The tensile strength and Young’s modulus of FRP were performed using the ASTM D 790-17 method, the sample size was width:thickness:length = 25 × 2 × 250 mm, and the instrument was performed using a tensile measuring instrument (manufacturer: Instron, Norwood, MA, USA, model: Instron 5982). To confirm the resin removed, the true density and porosity of the residue obtained from FRP pyrolysis were measured using a true density meter (manufacturer: Micromeritics, Norcross, GA, USA, model name: AccuPyc II 1340) as comparing with the density of a fiber yarn. For the analysis, approximately 2 g of the specimen was placed in a sample cup and then measured. The functional groups of the oxidized CF and CFRP and GFRP as a function of pyrolysis temperature were analyzed using high-performance x-ray photoelectron spectroscopy (XPS, manufacturer: Thermo, Waltham, MA, USA, model name: K-ALPHA+).

## 3. Results and Discussion

### 3.1. Tensile Strength and Young’s Modulus of FRP

The composites used in this study were prepared by a hand lay-up method, and the mechanical strengths of the composites differed depending on the type of fiber. CFRP, which uses carbon fiber with strong carbon–carbon bonds as reinforcement, showed a high tensile strength. Moreover, the Young’s modulus of the composites, which refers to the degree of resistance to elastic deformation under load [18], was found to be approximately twice that of glass-fiber-reinforced plastic (GFRP), since the strong carbon bonds are highly brittle in Table 2.

### 3.2. Surface Morphology of FRP According to Pyrolysis Temperatire

The decomposed morphology of FRP resin according to the pyrolysis temperature was confirmed through SEM in Table 3 and Table 4. As the thermal decomposition temperature of the resin used as the base material of the composite is in the range of 300–400 °C [15,16], the resin decomposition proceeded at a pyrolysis temperature of 400 °C and the fiber residue was clearly identified, regardless of the type of fiber.

At a pyrolysis temperature higher than 600 °C, the carbon fiber was burned and observed in a powder form. In the case of GF, the melting on the surface was observed at 600 °C, and as the melting increased to 800 °C, a tangled form was observed. In the case of BF, a tangled form was partially observed at 800 °C.

### 3.3. Resin Content Analysis of the Pyrolyzed Hand Lay-Up Composites

The composition content of resin and fiber were determined by analyzing the weight change during the pyrolysis of the composite using TGA under the conditions of a heating rate at 5 °C/min and temperature range of 30–800 °C in an air atmosphere in Figure 1. The composition of CFRP (CF-H Company) was confirmed to be 32.47 wt% resin and 65.47 wt% fiber. The thermal decomposition temperature of UPE, which is used as the base material of the composite, was 350 °C and progressed up to 540 °C, followed by complete removal. The thermal decomposition of the carbon fiber textile started at 524.25 °C and almost disappeared before the heat treatment temperature reached 800 °C.

To optimize the heat treatment conditions (temperature and time) under which the resin is removed without fiber heat loss, the resin pyrolysis temperature conditions were measured in 100 °C increments up to 800 °C. The heat treatment time was measured to be 240 min and the degree of thermal decomposition of the resin was continuously observed. In the case of CFRP, the decomposition of the resin mainly proceeded at 300 °C. Before the thermal decomposition temperature of the carbon fiber at the pyrolysis temperature of 400 ℃, most of the resin was removed. This was because there was almost no change in weight. However, at a pyrolysis temperature higher than 500 °C, the thermal decomposition of fibers occurred partially. As a decrease in the slope that does not result in weight change refers to the completion of thermal decomposition of the resin, the starting point of the time when there was almost no change in slope was identified by differentiating the TGA graph to optimize the heat treatment time. In the differential graph, the resin of the composite was completely removed at 8.39 min for a pyrolysis temperature of 400 °C and at 2.6 min for a pyrolysis temperature of 500 °C. Meanwhile, the resin and carbon fiber were decomposed at a pyrolysis temperature of 600 °C, and the completion time of the pyrolysis was confirmed to be 63.43 min. As the thermal decomposition of carbon fiber occurred simultaneously, an increase in the completion time of the pyrolysis was observed.

CFRP (CF-T Company) consisted of 35.62 wt% resin and 62.72 wt% fiber in Figure 2. In the differential graph, the resin of the composite was confirmed to be completely removed at 8.45 min for a pyrolysis temperature of 400 °C and at 2.54 min for a pyrolysis temperature of 500 °C. Meanwhile, the resin and carbon fiber were decomposed at a pyrolysis temperature of 600 °C, and the completion time of the pyrolysis was confirmed to be 82.04 min.

In commercial CFs, a difference in the degree of thermal decomposition was observed depending on the manufacturer. In the case of CF (T Company), the thermal decomposition of the fiber was not observed at 500 °C, but in the case of CF (H Company), it occurred at 500 °C. This phenomenon might be due to the difference in the carbonization temperature during the CF manufacturing process. The carbonization temperature for the CF (T Company) was 1500 ℃, whereas it was 1000 ℃ for the CF (H Company).

The fiber content of the GFRP, which was approximately 73 wt%, was slightly higher than that of CFRP, and the thermal decomposition of the glass fiber textile was barely observed in the temperature range of 800 °C in Figure 3.

In the differential graph, the resin of the composite was confirmed to be completely removed at 8.33 min for the pyrolysis temperature of 400 °C and 3.33 min for 500 °C. At a pyrolysis temperature of 600 °C, the resin was decomposed, and the completion time of the pyrolysis was confirmed to be 4.26 min. In the case of GF, the thermal decomposition was barely observed at 600 °C. Therefore, no increase in the completion time of pyrolysis was observed even at higher pyrolysis temperatures.

When continuous filament mat (CFM) was applied as reinforcement, the amount of resin impregnation increased with a composition of 33 wt% fiber and 66 wt% resin because considerable space existed between the filaments compared with the fiber textile type in Figure 4.

In the differential graph, the resin of the composite was confirmed to be completely removed at 8.37 min for the pyrolysis temperature of 400 °C and at 8.33 min for the pyrolysis temperature of 500 °C. At a pyrolysis temperature of 600 °C, the resin was decomposed, and the completion time of the pyrolysis was confirmed to be 4.11 min. These results show that the composite resin was removed within 10 min using TGA in an air atmosphere at the pyrolysis temperature of 400 °C.

### 3.4. True Density and Porosity of Fibers Obtained from FRP Analysis

The true density and porosity of the fiber-reinforced residue obtained after thermally decomposing 4 g or more of the composite specimens in a muffle furnace were measured and compared with that of a fiber yarn to confirm whether the thermal decomposition of the resin was well-performed. In addition, because the fiber used as reinforcement might be oxidized during the thermal decomposition process of the resin, the true density and porosity of the fiber subjected to thermal oxidation at 400 °C for 4 h in an air atmosphere were compared.

By introducing oxygen functional groups on the surface during the thermal oxidation of CF, GF, and BF [19], the true density of the oxidized fibers was increased by approximately 0.64–4% compared with that of untreated fibers (Table 5).

The determination of the true density of CFRP as a function of pyrolysis temperature revealed that the true density was low up to 200 ℃, which is the temperature before the resin with low density (density: 1.15 g/cm^3^) decomposed (Table 6). Then, a significant increase was observed from 300 °C, which is the temperature at which the resin decomposed. Subsequently, an increase in the true density and porosity was simultaneously observed owing to the formation of ash (density: 1.90 g/cm^3^) by the thermal decomposition of CF within the range of 500–600 °C or higher. The ash formation, due to the thermal decomposition of CF at a pyrolysis temperature higher than 600 °C, was also confirmed in the SEM analysis result. In the case of GFRP and GFRP (CFM), an increase in true density from 300 °C was also observed. Moreover, as the thermal decomposition of GF barely occurred, significant changes in the true density and porosity were not observed as a function of temperature. However, the true density and porosity decreased owing to the tangle because the GF melted at 800 °C or higher.

Significant changes in true density and porosity were also observed in the residue obtained from heat treatment using a muffle furnace because the resin removal started from a pyrolysis temperature of 300 °C. The true density and porosity were found to be stable at a pyrolysis temperature of 400 °C without the thermal decomposition and melting of fibers.

### 3.5. Functional Amounts of Fibers, Oxidized Fibers, and Fibers Obtained from FRP Analysis

Oxygen content of CF and GF was introduced by oxidation heat treatment reaction at 400 °C, and the oxygen content of each was observed to be increased to 3.48 and 36.79% of CF and GF compared with untreated fibers (Table 7). With the introduction of this oxygen content, a density increase of the oxidized fiber was observed.

When UPE was used as an impregnated resin, the O-C=O, C=O content of UPE affects the FRP element content. In the case of CFRP, the C content was increased, and the O content was decreased from 400 °C; conversely, in GFRP based on Si-O-Si, the C content was reduced from 400 °C (Figure 5). From the above results, it was thought that the decomposition of the impregnated resin of CFRP and GFRP was completely achieved at 400 °C. To confirm the above in detail, CFRP and GFRP C1s peaks according to the pyrolysis temperature were separated, functional group changes were confirmed, and the O-C=O, C=O content of UPE was completely reduced at the temperature where the change in element content was observed (Table 8).

## 4. Conclusions

In this study, TGA was employed to determine the resin content and optimize the heat treatment conditions (temperature and time) in which only the resin was removed without fiber heat loss. Accordingly, the measurement was performed every 100 °C at a resin pyrolysis temperature up to 800 °C with a heat treatment time of 4 h to continuously observe the thermal decomposition degree of the resin. The thermal decomposition of UPE was confirmed at the melting point (350 °C) regardless of the type of fiber used as reinforcement. In the case of CFRP, the resin decomposition mostly occurred at 300 °C. Notably, most of the resin was removed at a pyrolysis temperature of 400 °C, which is lower than the thermal decomposition temperature of carbon fiber, as almost no change in weight was observed. However, at a pyrolysis temperature of 500 °C or higher, the thermal decomposition of fibers partially occurred. The results showed that a pyrolysis temperature of 400 °C in an air atmosphere for less than 10 min was the optimal condition to analyze only the resin content without fiber loss. In addition, the true density and porosity of the fiber residues from the heat-treated FRP at each pyrolysis temperature by a muffle furnace were measured to confirm that only the resin was removed without fiber loss. Significant changes in the true density and porosity were also observed in the residue obtained from the heat treatment using a muffle furnace because the resin removal started from the pyrolysis temperature of 300 °C. The true density and porosity were found to be stable at a pyrolysis temperature of 400 °C without the thermal decomposition and melting of fibers.

## Figures and Tables

**Figure 1 materials-15-08182-f001:**
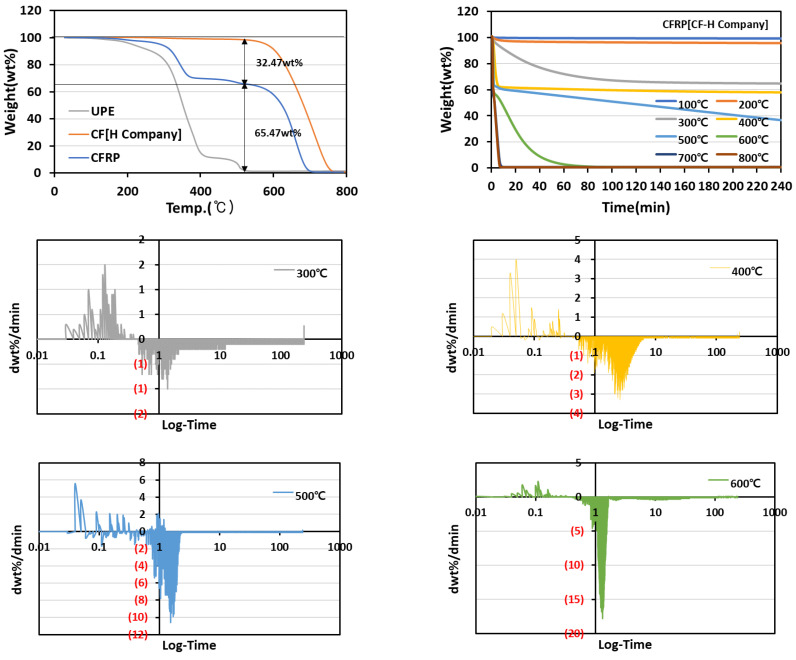
CFRP[CF-H Company] weight change and heat treatment time according to pyrolysis temperature.

**Figure 2 materials-15-08182-f002:**
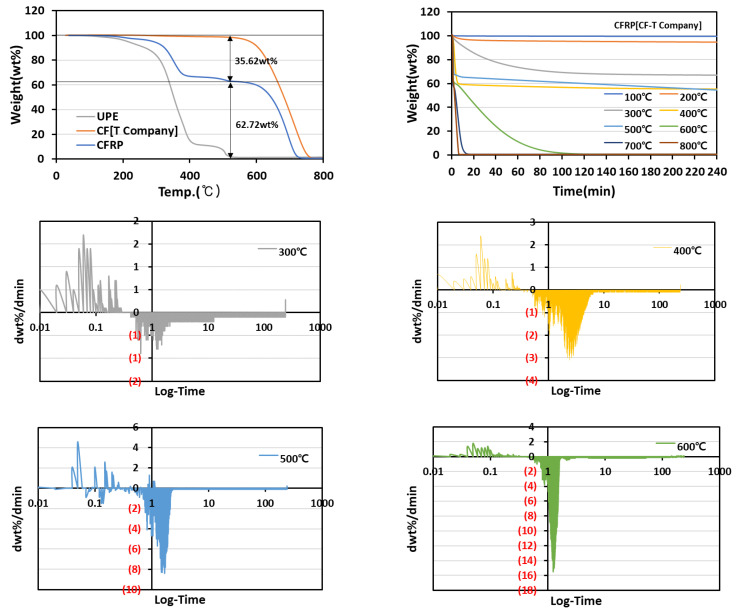
CFRP[CF-T Company] weight change and heat treatment time according to pyrolysis temperature.

**Figure 3 materials-15-08182-f003:**
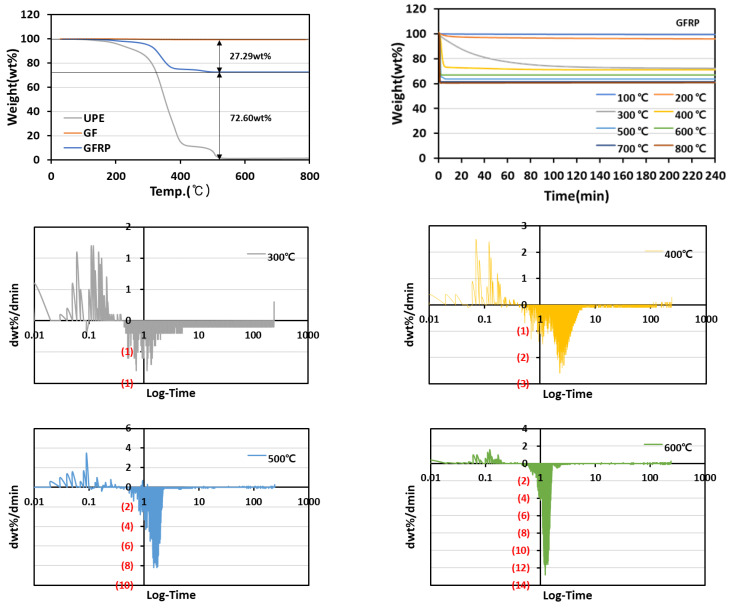
GFRP weight change and heat treatment time according to pyrolysis temperature.

**Figure 4 materials-15-08182-f004:**
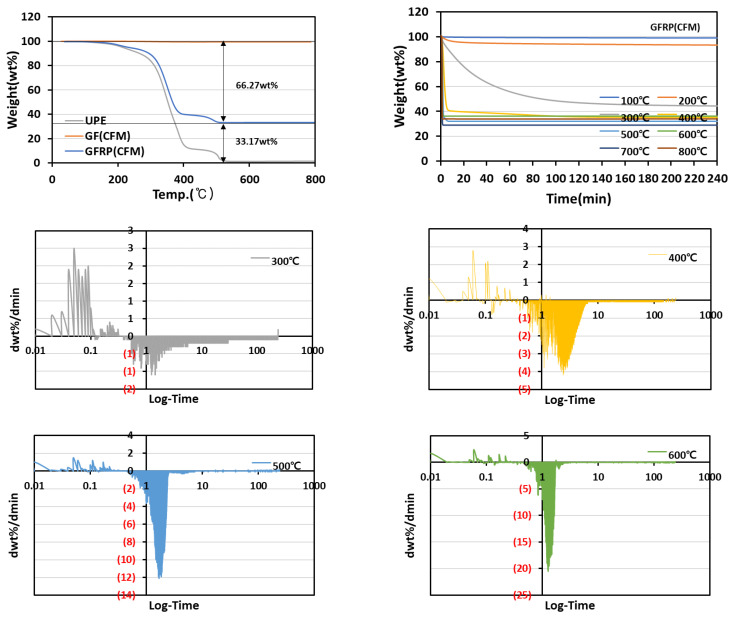
GFRP (CFM) weight change and heat treatment time according to pyrolysis temperature.

**Figure 5 materials-15-08182-f005:**
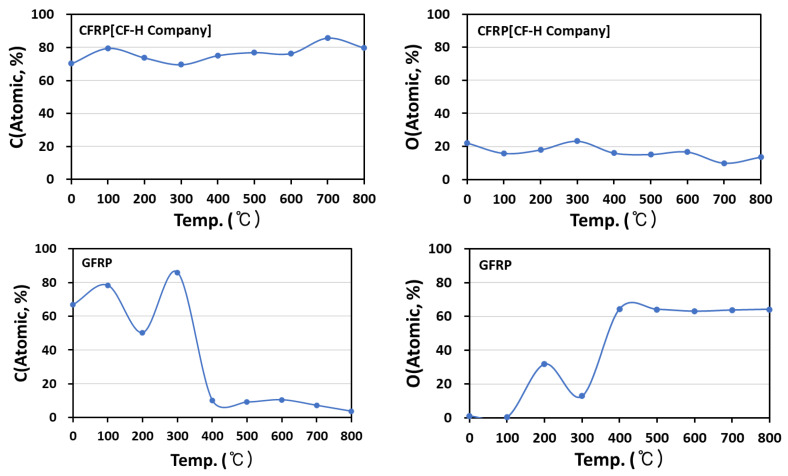
Functional amounts of fibers obtained from FRP analysis according to pyrolysis temperature.

**Table 1 materials-15-08182-t001:** Mechanical properties of FRP (article data).

Mechanical Properties	Unit	CFRP	GFRP	AFRP	BFRP
Density	g/m^3^	1.5–2.1	1.25–2.5	1.25–1.45	2.65
Tensile strength	GPa	0.6–4.0	0.483–4.58	1.7–3.6	2.9–3.1
Young’s modulus	GPa	37–784	35–86	41–175	65–75
Elongation break	%	0.5–1.8	1.2–5	1.4–4.4	3.15

**Table 2 materials-15-08182-t002:** Tensile strength and Young’s modulus of the hand lay-up molded FRP.

Composites	Tensile Strength	Tensile Modulus	Young’s Modulus	Elongation Break
Unit	GPa	GPa	GPa	%
CFRP[CF-H Company]	0.70	49.36	30.12	2.31
CFRP[CF-T Company]	0.67	49.42	29.12	2.01
GFRP[OCM]	0.38	21.96	16.32	1.99
GFRP[CFM]	0.09	8.92	3.83	1.39
AFRP	0.26	19.56	18.99	1.37

**Table 3 materials-15-08182-t003:** Surface morphology of CFRP, GFRP, and BFRP according to pyrolysis temperature (30×).

Pyrolysis Temperature (°C)	CFRP	GFRP	BFRP
	30×		
100	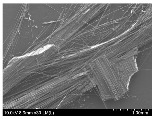	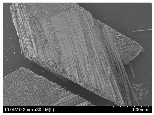	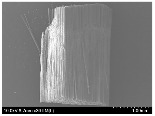
200	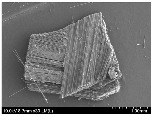	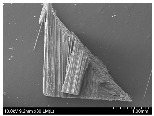	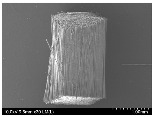
300	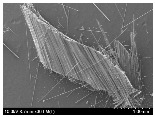	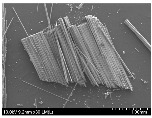	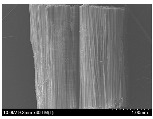
400	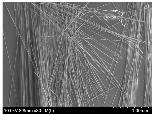	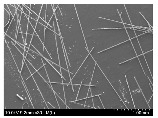	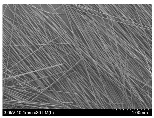
500	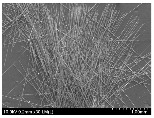	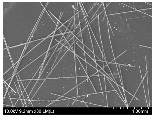	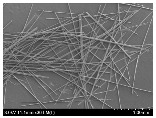
600	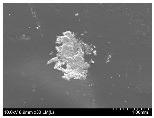	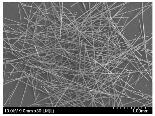	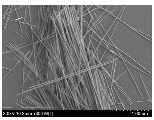
700	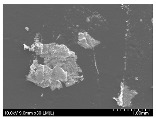	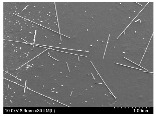	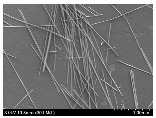
800	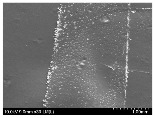	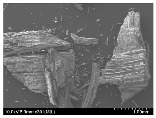	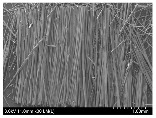

**Table 4 materials-15-08182-t004:** Surface morphology of CFRP, GFRP, and BFRP according to pyrolysis temperature (1000×).

Pyrolysis Temperature (°C)	CFRP	GFRP	BFRP
	1000×		
100	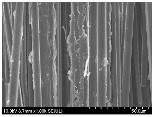	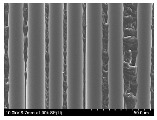	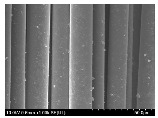
200	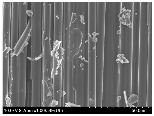	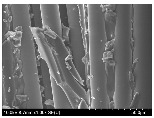	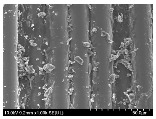
300	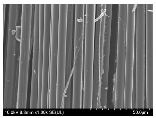	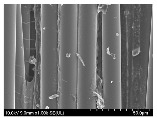	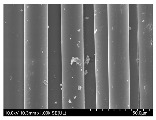
400	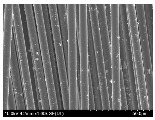	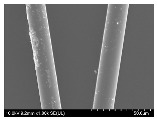	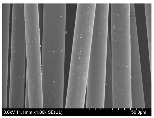
500	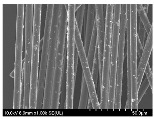		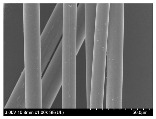
600	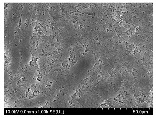	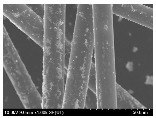	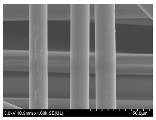
700	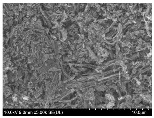		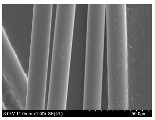
800	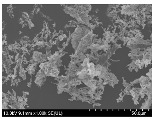	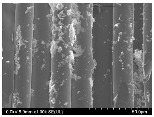	

**Table 5 materials-15-08182-t005:** True density and porosity of the fibers and oxidized fibers.

Sample			<Air Treatments> [Air, 400 °C, for 4 h]	
	True Density	Porosity	True Density	Porosity
	g/cm^3^	%	g/cm^3^	%
CF[H Company]	1.788	44.080	1.863	46.320
CF[T Company]	1.788	44.060	1.858	46.170
GF	2.624	61.880	2.641	62.140
BF	2.588	61.350	2.648	62.230

**Table 6 materials-15-08182-t006:** True density and porosity of FRP according to pyrolysis temperature.

Pyrolysis Temperature	CFRP [CF-H Company]		CFRP [CF-T Company]		GFRP		GFRP (CFM)	
	True Density	Porosity	True Density	Porosity	True Density	Porosity	True Density	Porosity
°C	g/cm^3^	%	g/cm^3^	%	g/cm^3^	%	g/cm^3^	%
-	1.473	32.130	1.493	33.010	1.820	45.050	1.483	32.570
100	1.518	34.120	1.504	33.490	1.839	45.620	1.479	32.360
200	1.485	32.660	1.508	33.680	1.979	49.470	1.489	32.840
300	1.989	49.730	1.823	45.160	2.599	61.520	2.624	61.890
400	1.910	47.840	1.822	45.130	2.647	62.220	2.649	62.250
500	1.962	49.040	1.811	44.790	2.655	62.330	2.668	62.510
600	1.953	48.800	1.880	46.820	2.673	62.590	2.674	62.600
700	1.944	48.570	1.897	47.290	2.661	62.410	2.675	62.610
800	1.949	48.700	1.888	47.060	2.268	55.910	2.623	61.880

**Table 7 materials-15-08182-t007:** Functional amounts of fibers and oxidized fibers.

Sample	C	O	N	Si	Sum
	Atomic %				
CF[H Company]	77.66	18.69	1.90	1.75	100.00
Oxidized CF	65.72	22.17	6.56	5.55	100.00
GF	79.59	18.80	0.73	0.88	100.00
Oxidized GF	21.04	55.59	0.42	22.95	100.00

**Table 8 materials-15-08182-t008:** Functional groups of fibers obtained from FRP analysis according to pyrolysis temperature.

Pyrolysis Temperature (°C)	CFRP			GFRP		
	C-C	C-O	C=O, O-C=O	C-C	C-O	C=O, O-C=O
	284.5	286.3	288.6	284.5	286.3	288.6
	Content (%)					
-	80	16	4	80	14	6
100	86	12	2	82	13	4
200	83	15	3	76	17	7
300	79	16	3	83	14	3
400	65	35	-	-	100	-
500	85	15	-	-	100	-
600	82	18	-	-	100	-
700	88	12	-	-	100	-
800	79	21	-	-	100	-

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
