# Peer review of "Optimizing Heat Treatment Conditions for Measuring CFRP and GFRP Resin Impregnation"

_materials, 2022, doi:10.3390/ma15228182_

Round 1

Reviewer 1 Report

  The manuscript is about determining the content of the unsaturated polyester resin in carbon fiber and glass fiber composites by thermal analysis.

My comments:

- Please explain what "melting temperature" means (rows 15, 152, 259), knowing that unsaturated polyester resins are thermoset polymers. You cannot talk about the melting temperature because they cannot be made liquid again by increasing the temperature, as can thermoplastic polymers.

- Table 1 is not related to the text of the manuscript. In Table 1 are the manufacturer specifications or the authors' determinations? CFRP from sheet 1 are from manufacturer H or T?

- Table 2 -  there are no determinations for BFRP or AFRP.

- Chapter 2.3 - The analysis method and the dimensions of the samples are presented, but the device (model, year of manufacture) with which the tensile strength and young's modulus of FRP were determined is not shown.

 I recommend publishing the manuscript only after clarifying the above.

Author Response

Q1) Please explain what "melting temperature" means (rows 15, 152, 259), knowing that unsaturated polyester resins are thermoset polymers. You cannot talk about the melting temperature because they cannot be made liquid again by increasing the temperature, as can thermoplastic polymers.

A1) UPE was used as a thermosetting polymer, so the word melting point was changed to the pyrolysis temperature.

Q2) Table 1 is not related to the text of the manuscript. In Table 1 are the manufacturer specifications or the authors' determinations? CFRP from sheet 1 are from manufacturer H or T?

A2) Table 1 was a reference to the paper data, and Table 1 was displayed in the text, and the references were also shown. The manufacturers of carbon fibers used in CFRP were distinguished, and the manufacturers of carbon fibers were represented by Hyosung as H and Toray as T.

Q3) Table 2 -  there are no determinations for BFRP or AFRP.

A3) AFRP was molded by the hand layup method, and the physical property data of the composite was added to Table 2. BFRP was unable to add data to this paper due to the difficulty of obtaining textiles.

Q4) Chapter 2.3 - The analysis method and the dimensions of the samples are presented, but the device (model, year of manufacture) with which the tensile strength and young's modulus of FRP were determined is not shown.

A4) The tensile strength of FRP and the measurement of young's modulus were performed using ASTM D790-17 method, the sample size was width: thickness: length = 25x2x250 mm, and the instrument was performed using a tensile measuring instrument (manufacturer: Instron, model: Instron5982).

Reviewer 2 Report

An interesting manuscript, please consider the following points to improve the quality of your manuscript:

1. ''The impregnated resin was prepared by mixing the primary material, hardener, 99 and wax in a ratio of 300:6:3 g.'' Please mention about the mixing method; the temperature, time and stirring rate. Did you mix altogether or one by one?

2. ''composite thickness of 2 mm'' how you measured it? please mention it.

3. You need to check the FT-IR confirm the structure. 

Author Response

Q1) ''The impregnated resin was prepared by mixing the primary material, hardener, 99 and wax in a ratio of 300:6:3 g.'' Please mention about the mixing method; the temperature, time and stirring rate. Did you mix altogether or one by one?

A1) To mix the material, the material was put into a 1 L plastic beaker at once and mixed with a stirring speed of 100-150 rpm using a mechanical stirrer at room temperature.

Q2) ''composite thickness of 2 mm'' how you measured it? please mention it.

A2) The thickness of the composite was measured using a digital vernier caliper.

Q3) You need to check the FT-IR confirm the structure. 

A3) In order to observe in detail the elemental content and functional group morphology changes of CFRP and GFRP according to the pyrolysis temperature, XPS was used for analysis, and the results were added to Article 3.5.

Round 2

Reviewer 2 Report

Please include the FT-IR analysis, it will confirm the modified structure of material.

Author Response

Q1) Please include the FT-IR analysis, it will confirm the modified structure of material.

A1) The reason why XPS was used instead of FT-IR for carbon fiber functional group analysis was that carbon fiber had a very low functional group peak intensity unlike glass fiber, making it difficult to distinguish the functional group content. There was a reference paper that the FTIR functional group peak strength was affected by the heat treatment temperature of carbon fiber, and we will check the exact reason in detail.